# Differential Response to Single and Combined Salt and Heat Stresses: Impact on Accumulation of Proteins and Metabolites in Dead Pericarps of *Brassica juncea*

**DOI:** 10.3390/ijms22137076

**Published:** 2021-06-30

**Authors:** Jeevan R. Singiri, Bupur Swetha, Noga Sikron-Persi, Gideon Grafi

**Affiliations:** French Associates Institute for Agriculture and Biotechnology of Drylands, Jacob Blaustein Institutes for Desert Research, Ben-Gurion University of the Negev, Midreshet Ben Gurion 84990, Israel; jeevannaveen01@gmail.com (J.R.S.); bupurswethanagaraju1234@gmail.com (B.S.); sikron@exchange.bgu.ac.il (N.S.-P.)

**Keywords:** dead pericarps, salinity, short episodes of high temperature, stress response, reproductive phase, seed abortion, phytohormones, *Brassica juncea*

## Abstract

Dead organs enclosing embryos, such as seed coats and pericarps, are emerging as important maternally-derived components of the dispersal unit that affect seed performance and fate. In the face of climate change and increased incidents of heatwaves, we sought to investigate the effect of salinity (S), short episodes of high temperature (HS), and combination of S + HS (SHS), at the reproductive phase, on the properties of dead pericarps of *Brassica juncea*. Proteome and metabolome analyses revealed multiple proteins and metabolites stored in dead pericarps whose levels and composition were altered under single and combined stress conditions. The protein profile of SHS showed a higher correlation with salt than with HS indicating the dominant effect of salt over heat stress. On the other hand, the analysis of metabolites showed that the profile of SHS has better correlation with HS than with salt. The integration of metabolic and proteomic data showed that changes in TCA cycle intermediates and certain amino acids (e.g., proline) under salt treatments (S and SHS) are highly correlated with changes in proteins involved in their biosynthetic pathways. Thus, accumulation of proteins and metabolites in dead pericarps is differently affected by single and combination of salt and heat stresses. Salinity appears to dominate plant response to combined stresses at the protein level, while heat appears to be the major factor affecting metabolite accumulation in dead pericarps.

## 1. Introduction

In virtually all agricultural regions, abiotic stresses such as drought, salinity, and temperature extremes reduce average yields for most major crop plants by more than 50%, presenting a huge barrier to feeding an ever-growing world population [1,2,3]. With the expected changes in global climate, environmental stresses are likely to increase in severity leading to serious effects on crop yields [4]. Moreover, soil salinization affects an estimated 1 to 3 million hectares in the enlarged EU and is considered a major degradation process endangering the potential use of European soils [5]. The average global annual temperature has increased by 1.1 °C compared to the average temperature at the preindustrial era, and it is predicted to keep increasing by 3–5 °C by the end of this century, due to accumulation of greenhouse gases in the atmosphere. Yet, climate change not only impacts the average annual temperature, but also the frequency of incidents of extreme climate events, including heatwaves and hot spells [6,7,8]. Hot spells are the most critical factor affecting crop yield particularly when appear in combination with other stresses and during flowering and seed development [8,9]. Thus, exposure of mother plants to stress conditions during vegetative and reproductive stages has a great impact on progeny seed and dispersal unit properties [10,11,12].

The unique response of plants to a combination of stresses has been noted previously [13] and since then was studied intensively [14,15,16]. The emerging theme suggests that exposure of plants to combination of abiotic stresses or abiotic and biotic stresses, particularly during the reproductive stage culminates in a unique response that has a great negative impact on yield [8,17]. Maternal environment not only affecting the embryo properties but also the properties of the maternally derived dead organs enclosing the embryo (DOEE) [18,19]. DOEEs appear to be important components of the dispersal unit (DU) that have been evolved in plants probably in conjunction with their habitats to carry out multiple functions, that is to nurture the embryos and to ensure offspring success in their ecological niche [18,20]. Thus, besides providing a protective shield for the embryo and dispersal accessories, DOEEs also function as a long-term storage for proteins such as hydrolases, reactive oxygen species (ROS) detoxifying enzymes and cell wall modifying enzymes as well as regulatory substances that control plant growth and development, microbial growth, as well as germination of heterologous species [19,21,22,23,24]. The expected changes in DOEEs properties because of maternal environment could have an impact on seed viability and persistence, germination, and seedling establishment and consequently on plant population dynamics and diversity [25,26].

Crop plants are essentially highly sensitive to abiotic stresses that cause significant yield losses worldwide [3,27]. Multiple studies related to the effect of heat shock on plant performance were performed under long-term exposure (often >24 h) to high temperatures (37–45 °C), though in recent years the effect of short episodes of high temperature (heatwaves/hot spells) are getting more attention [8,28]. Here, we sought to examine the effect of single and combined salinity and short-term exposure to high temperature during the reproductive phase on DOEE properties of *Brassica juncea* (L.) Czern & Coss. (Brassicaceae) that together with other *Brassica* species represent an important source of vegetable oil worldwide [29,30]. Like other *Brassica* and leguminous crop plants, *B. juncea* fruits are indehiscent, that is, the fruit remains intact and is not splitting open at maturity. We performed proteomics and metabolomics, to investigate the effect of maternal environment on the properties of dead pericarps. Our data demonstrate unique responses of plants to single and combined salinity and heat stresses. Salinity appears to dominant plant response to combined stresses at the protein level, while heat appears to be the chief factor affecting metabolites accumulation in dead pericarps of *B. juncea*.

## 2. Materials and Methods

### 2.1. Plant Growth Conditions and Exposure to Stress

*Brassica juncea* (Indian mustard) seeds purchased from the local market were sown in standard gardening soil composed of peat and perlite (2:1 ratio) in small pots. Mustard seedlings were transplanted (at 18 days after sowing) into 1L pots having red sandy soil (Hamra) [31] supplemented with 4 g/L slow release fertilizer (Green Multigan 20% N, 11% P_2_O_5_, 16% K_2_O and trace elements). Briefly, plants were irrigated every two days with tap water for one month until the beginning of bolting. At which time half of the plants were exposed to salt stress of 50 mM NaCl for one month after which NaCl concentration was gradually increased, in a 5-day manner, to 75, 100, 150, up to 200 mM. After reaching the highest salt concentration, half of the water and salt irrigated-plants were subjected to 3 intervals of heat shock treatment (37 °C, 3 h each) in a course of 5 days to obtain a moderate effect of heatwave. According to the World Meteorological Organization a heatwave is defined as 5 or more consecutive days of prolonged heat in which the daily maximum temperature is higher than the average maximum temperature by 5 °C (9 °F) or more (https://www.encyclopedia.com/environment/energy-government-and-defense-magazines/heat-waves accessed on 11 May 2021). Notably, at the time of salt application plants were irrigated with distilled water (DW) or DW + salt. The irrigation with 200 mM NaCl was continued for another week and then all pots were irrigated with tap water until fruit matured and dried out, and the mustard pods were harvested for further analysis.

### 2.2. Proteome Analysis

For proteome analysis, 10 mg of ground pericarps derived from control and stress-treated plants was placed in 2 mL tube with 100 μL PBS and incubated at 4 °C for 1 h with gentle rotation, then centrifuged at 4 °C at high speed (16,000× *g*) for 10 min. Fifty microliters of supernatants were collected, lyophilized, and stored at −20 °C until used for comparative, quantitative proteome analysis.

Proteome analysis of pericarps (three replicates) was performed by the proteomic services of The Smoler Protein Research Center at the Technion, Israel using LC-MS/MS on LTQ Orbitrap (ThermoFisher Scientific, Waltham, MA, USA; https://proteomics.net.technion.ac.il/proteomic-services/ accessed on 11 May 2021), followed by identification and quantification by MaxQuant, using *Brassica rapa subsp. pekinesis* proteins from UniProt as a reference. Note that each replicate contains randomly selected pericarps from a pool of 16 plants. Quantification and normalization were performed using the LFQ method. Subsequent bioinformatic analysis was carried out at the Bioinformatics Core Facility, Ben-Gurion University, using commercial software Partek Genomics Suite (version 6.6, 2015 Partek Inc., St. Louis, MO, USA) as well as a leading publicly available software, the R package (http://www.R-project.org/ accessed on 11 May 2021). Proteins marked as “contaminant”, “reverse” and “only identified by site” were filtered out. In an additional filtering step, only proteins in which at least one of the groups had at least 2 non-zero replicates were retained. LFQ intensities were Log_2_ transformed, and zero intensities were imputed (replaced) by random numbers derived from a normal distribution in the low expression range (width = 0.2, downshift = 1.6). Imputation was repeated 10 times to avoid relying too heavily on fabricated numbers. Each of the 10 imputed datasets was submitted to hypothesis testing for differential protein expression using Limma [32]. The statistical model tested the contrast between salt proteins and control proteins. A protein was considered differentially expressed (DE) if it had nominal *p*-value < 0.05 and absolute fold change (in linear scale) >1.3 in at least 8 of the 10 imputed datasets. Subsequently, a more stringent cutoff was used, requiring an FDR-adjusted *p*-value < 0.05 in at least 8 of the 10 imputed datasets.

### 2.3. Primary Metabolites Analysis

Quantification of primary metabolites were performed using GC–MS method essentially as described [33]. Briefly, powdered lyophilized pericarps (70 mg) were extracted with 1 mL of a precooled mix containing methanol, chloroform and MiliQ water (2.5:1:1 *v*/*v*, respectively) supplemented with ribitol as the internal standard (4.5 μg/mL) and vortexed thoroughly. Following incubation for 10 min at 25 °C on an orbital shaker, samples were sonicated for 10 min in ultra-sonication bath at room temperature and centrifuged at high speed (10 min, 16,000× *g*). The supernatant was collected, added 300 µL of miliQ water and 300 µL of chloroform, vortexed for 10 s and centrifuged for 5 min at high speed and the upper phase was collected and aliquots were lyophilized and subjected to derivatization. Derivatization was performed by adding 40 µL methoxyamine hydrochloride (20 mg/mL in pyridine) to the dry sample and incubation for 2 h at 37 °C on a shaker platform. Samples were added 70 µL MSTFA and 7 µL of alkane mix and incubated with constant shaking for 30 min at 37 °C. Samples were subjected for gas chromatography- mass spectrometry (GC-MS) analysis (Agilent Ltd., Santa Clara, CA, USA) as described in [33,34]. Separation was carried out on a Thermo Scientific DSQ II GC/MS using a FactorFour Capillary VF-5ms column (Agilent Ltd, Santa Clara, CA, USA). Acquired chromatograms and mass spectra were evaluated using Xcalibur (version 2.0.7) software and metabolites were identified and annotated using the Mass Spectral and Retention Time Index libraries available from the Max-Planck Institute for Plant Physiology, Golm, Germany (http://csbdb.mpimp-golm.mpg.de/csbdb/gmd/msri/gmd_msri.html accessed on 11 May 2021). Data were log2 transformed and QA plots were prepared using Partek Genomics Suite. Further analysis was carried out after scaling normalization, by shifting the log2 intensities to the grand mean. Statistical testing for differential abundance by 1-Way ANOVA for the treatment effect was performed in Partek Genomics Suite. Cutoff for differential abundance: unadjusted *p*-value < 0.05 and fold change (in linear scale, either direction) >1.5. The level of metabolites was calculated by normalizing the intensity of the peak of each metabolite to grand mean. PCA, ANOVA, Student’s *t* tests, and hierarchical clustering analysis were performed using the Metaboanalyst 4.0 [35].

## 3. Results

### 3.1. Effect of Maternal Environment on Protein Accumulation in Dead Pericarps: Proteome Analysis

Proteins derived from pericarps of control and stress-treated plants were subjected to proteome analysis. After filtering out potential contaminant, reverse, only identified by site as well as filtering for proteins expressed in at least two replicates of at least one treatment group, 1620 proteins were documented (Appendix A). Protein class categorization showed that among the 1078 proteins identified in this category, and 558 and 118 are related to metabolite and protein modifying enzymes, respectively (Figure 1A). Metabolite modifying enzymes include hydrolases (124 proteins) and oxidoreductases (230 protein (Figure 1B). Interestingly, among the 118 protein modifying enzymes, 114 are classified as proteases (Figure 1C), which might be involved in the execution of programmed cell death of the pericarps or being accumulated in dead pericarps for regulating multiple processes including embryo protection from potential soil pathogens and germination [36,37]. A principal component analysis (PCA) (Figure 2A) of the LFQ values separated all treatments with PC1 explaining 33.4% of the variance essentially separating salt treatments (S and SHS) from control and HS, while PC2 (17.1% of the variance) separated HS treatments from control and salt treatments. These demonstrate the unique effect of salt and HS on proteins accumulated in dead pericarps. Among the proteins identified, 1248 proteins that pass the cutoff of FDR adjusted *p*-value < 0.05 in any of the pairwise comparisons were differentially accumulated in pericarps (Appendix A). Sample-wise correlation of the differentially present (DP) proteins between treatments revealed that combined SHS is better correlated with salt (*r* = 0.65) than with HS (*r* = 0.44) (Figure 2B), implying that under combined stresses, the response to salt dominates the response to HS. Hierarchical clustering of differentially present proteins further demonstrates the similarity in protein profile between salt and SHS (Figure 2C).

Setting the cut-offs for differentially present proteins to FC > 2 between treatments and control we found 562, 367 and 209 DP proteins for SHS, salt and HS, respectively, that were up-accumulated in comparison to control (Appendix A). Venn diagram showed (Figure 2D) that SHS pericarps possess 244 unique proteins and share 273 proteins with salt and 78 proteins with HS pericarps. Most notable is the very high accumulation of thionin and Knot1/defensin (124 and 63 FC SHS/C, respectively) in SHS pericarps but not in pericarps derived from salt and HS-treated plants. Proteins up-accumulated in salt but not in HS and SHS pericarps include sinigrinase/myrosinase (11-fold) involved in plant defense against herbivores and pathogens [39], as well as FK506 binding protein (10-fold), a member of peptidyl-prolyl cis-trans isomerase (PPIase) family that catalyzes the interconversion between prolyl cis/trans conformations and acting as a molecular switch involved in diverse processes essential for plant growth and development and response to stress [40]. Among the proteins affected mostly by heat stress and were up-accumulated in HS and SHS pericarps are heat shock proteins including HSP70-5 (38-fold), HSP104 (31-fold), as well as multiple small HSPs. Furthermore, osmotin-like protein that belongs to pathogenesis-related 5 (PR-5) group of proteins as well as Kunitz trypsin inhibitor 5 (KTI5, 76-fold) were up-accumulated.

Categorization for biological process of proteins up-accumulated in pericarps derived from stress-treated plants compared to control plants revealed (Appendix A) a notable variation in accumulation of stimulus-responsive proteins. Accordingly, 20, 25, and 38 proteins were up-accumulated in salt, HS and SHS pericarps, respectively, compared to control. Salt pericarps share 13 proteins (65%) with SHS, 4 (20%) proteins with HS pericarps, while HS pericarps share 9 (36%) proteins with SHS pericarps. Among the protein in this category, three proteins were common to all pericarps, namely, Cell Division Control Protein 48 homolog E, Protein disulfide-isomerase A3 and Calcium-binding protein CML9-related. Furthermore, pericarps derived from HS and SHS-treated plants vs. control showed up-accumulation of small HSPs (SHSPs), while none SHSP was up-accumulated in salt vs. control pericarps (Appendix A).

Finally, molecular class categorization of DP proteins revealed that under single and combined stresses metabolite modifying enzyme was the major group up-accumulated in pericarps derived from stress-treated plants compared to control, demonstrating 33.5, 35 and 29.9% of DP proteins in salt, HS and SHS, respectively (Appendix A). Also notable is the up-accumulation of protein modifying enzymes, most of which are proteases, in dead pericarps (Appendix A). Thus, we identified 20, 13 and 26 proteases in salt, HS and SHS pericarps, respectively. Interestingly, metalloproteases were abundant in pericarps of salt and SHS, while cysteine proteases were particularly up-accumulated in HS; serine proteases were common to all treatments (Appendix A).

### 3.2. Single and Combined Stresses Alter Primary Metabolites Accumulated in Dead Pericarps

We performed metabolite profiling of pericarps derived from control and stress-treated plants of *B. juncea* using GCMS and identified 65 primary metabolites (Appendix A). A principal component analysis (PCA) of all identified primary metabolites showed that stress has a significant effect on the metabolites accumulated in dead pericarps (Figure 3A). Accordingly, the first principal component (PC1) demonstrates 38.8% of the variance separating all treatments, while PC2 accounting for 25.5% of the variance separated HS-treated samples from control and salt. Among the metabolites most affecting the separation of samples on PC1 were (in a decreasing order) ornithine, proline, lysine, alanine, serine, tyrosine and leucine (based on eigenvector values), while the metabolites most contributing to the variance on PC2 were cellobiose, isomaltose and rhamnose. Sample-wise correlation of DP metabolites between treatments showed that the metabolic profile of combined SHS has a higher correlation with HS (*r* = 0.77) than with Salt (*r* = 0.60) (Figure 3B). Hierarchical clustering analysis further showed that at the metabolite level single and combined stresses appear to be distinct from each other (Figure 3C). The relative content of multiple amino acids was significantly altered under salt treatment with most notable increase in proline and alanine and decrease in leucine (Table 1). The TCA cycle intermediates, malic acid, citric acid, fumaric acid, and succinic acid were down-accumulated in pericarps derived from stress-treated plants (Table 1). Multiple sugars were specifically up-accumulated in pericarps of HS-treated plants (HS and SHS) including cellobiose and isomaltose, while sorbitol was particularly up-accumulated in pericarps of salt-treated plants (salt and SHS) (Table 1).

### 3.3. Correlation between Metabolites and Proteins

Among the DP protein up-accumulated in salt pericarps is the enzyme Delta-1-pyrroline-5-carboxylate synthase A (P5CSA) that plays a key role in proline biosynthesis and its accumulation in pericarps (S/C FC = 58.9) is associated with the accumulation of proline to a very high level (S/C FC = 74) (Figure 4). Indeed, careful analysis of the proteome and metabolome data revealed that multiple metabolites whose abundance was altered in salt and SHS pericarps were highly correlated with the alteration of proteins/enzymes in their respective biosynthetic pathway (Table 2). For example, reduction in the TCA cycle intermediates citrate, fumarate, and malate is correlated with up-accumulation of aconitase that catalyzes the stereo-specific isomerization of citrate to isocitrate, fumarate hydratase that catalyzes the hydration of fumarate to malate and with malate dehydrogenase that catalyzes the conversion of malate to oxaloacetate (Table 2).

## 4. Discussion

We described here the considerable impact of single and combined salinity and short-term exposure to heat stress during the reproductive phase of the crop plant *B. juncea* on dead pericarp properties. Although an increase in global mean temperature by one degree Celsius is predicted to reduce significantly the yields of major crops such as wheat, rice, and maize [41], most detrimental effect on yield is exposure of crops during the reproductive stage to short episodes of high temperature (heatwave/warm spell) [42,43,44,45,46]. Thus, heatwaves and warm spells that are predicted to increase in frequency in many regions of the world [6,47,48] pose a serious threat to global food production.

Consistent with the idea that a combination of stresses is interpreted by plants as a unique stress, which induces distinctive plant response [13], our data demonstrate the peculiar response of mustard plants to a combination of salinity and short episodes of high temperature, which is not merely the sum up responses of the plant to heat and salt. Accordingly, we found that single and combined salinity and heat stress differently affect the accumulation of proteins and metabolites in dead pericarps of *B. juncea.* The present data are consistent with recently published data demonstrating the impact of abiotic stresses on the dispersal unit properties, particularly the properties of the DOEE of wild plants such as *Anastatica hierochuntica* and *Avena sterilis* [18,19,21,22]. Thus, the composition and level of proteins accumulated in dead pericarps *of A. hierochuntica* was altered under salt treatment with a notable up-accumulation of proteins related to response to biotic and abiotic stresses [21]. Likewise, the composition of metabolites and proteins in husks of natural populations of *Avena sterilis* are affected by rainfall [22]. The accumulation of substances in DOEEs appeared to be stress-specific. Accordingly, husks derived from *Avena* plants growing under drought conditions (155–180 mm) accumulated higher level of ABA compared to its accumulation in husks of plants growing under high precipitation regime (727–675 mm). On the other hand, exposure to salt resulted in down-accumulation of ABA in *B. juncea* pericarps (Swetha et al., 2021) and in the pericarp of *A. hierochuntica* [21].

Interestingly, the analysis of the proteome data revealed that the response of plants to combined salinity and heat stresses is dominated by salinity inasmuch as SHS protein profile showed a better correlation with salt (*r* = 0.65) than with heat (*r* = 0.44). Proteins accumulated in dead pericarps include metabolite modification enzymes involved in glycolysis, TCA cycle and amino acid biosynthesis, as well as hydrolases (e.g., nucleases, chitinases, and proteases), ROS metabolizing and cell wall modifying enzymes, proteases as well as multiple chaperones and chaperonins.

Although, the general proteome picture illustrates high correlation between the combined SHS stresses and salinity, detailed analysis of the data showed high correlation between SHS and HS with respect to heat shock proteins (chaperones), particularly small HSPs, which were accumulated in HS and SHS but not in salt pericarps. The expression of small HSPs is commonly induced under various stress conditions but particularly following exposure to high temperature [49]. These proteins are accumulated in petioles of *Zygophyllum dumosum* during the dry season [50] or in husks of *Avena sterilis* growing in natural habitats experiencing drought [22] and implicated in stress tolerance, at least partly by serving as molecular chaperones that stabilize and maintain proper protein folding [51,52].

Most intriguing is the up-accumulation of biotic defensive proteins in pericarps in response to abiotic stresses. The small proteins thionin and Knot1/defensin were remarkably up-accumulated only in pericarps derived from combined SHS-treated plants. These small proteins of about 5 kDa are toxic to plant pathogenic bacteria and fungi and might be involved in plant defense against biotic stresses [53,54]. Indeed, Defensin-like protein 195 was the only upregulated protein released from seeds of *A. hierochuntica* plants subjected to salt and combined salt and heat stresses. While defensins are induced in response to pathogen attack [55], multiple plants subjected to various abiotic stresses, including salt, drought, and cold showed the upregulation of defensins [56,57]. Thus, abiotic stresses elicit a response that prime the plant as well as its progeny seeds against potential pathogen attack that is mediated, at least partly, by small molecules defensins. The accumulation of these proteins in dead pericarps of *B. juncea* subjected to SHS represents another layer of protection of germinating seeds that could improve resistance to potential soil pathogen. Indeed, exposure of plants to abiotic stresses including wounding, salinity, heat and drought activates immunity response that confers better resistance to pathogens [58,59,60].

Similarly, salinity induces the accumulation in dead pericarps of sinigrinase/myrosinase (11-fold), which is involved in plant defense against herbivores and pathogens [39]. The myrosinase pathway, widely distributed in Brassicales, catalyzes the hydrolysis of glucosinolates into glucose and thiol that is converted rapidly to isothiocyanate or other less toxic molecules, which often act to deter herbivores [61].

Molecular class categorization revealed the up-accumulation in dead pericarps of protein modifying enzymes, most of which are proteases. Thus, we identified 20, 26, and 13 proteases in salt, SHS, and HS pericarps, respectively. Interestingly, while serine proteases were found in all treatments, metalloproteases were most abundant in pericarps of salt and SHS, while cysteine proteases were particularly up-accumulated in HS pericarps (Appendix A). Cysteine proteases were extensively studied and found to be involved in multiple processes in plants including senescence and programmed cell death as well as in processing and degrading seed storage proteins [62].

Among the proteins affected mostly by heat stress and were up-accumulated in HS and SHS pericarps are heat shock proteins including HSP70-5 (38-fold), HSP104 (31-fold), as well as multiple small HSPs (sHSPs), which are implicated in conferring multiple stress tolerance [63,64]; AtHSP101 can complement thermotolerance defect in *Saccharomyces cerevisiae* resulted from deletion of the HSP104 [65].

While at the proteome level, salinity appears to dominate plant response to combination of salinity and heat, at the metabolome level, heat appears to be the major factor influencing the response to a combination of stresses. Accordingly, we found a relatively higher correlation in metabolic profiles between SHS and HS (*r* = 0.77) than between SHS and salt (*r* = 0.6). Furthermore, sample-wise correlation of metabolites showed low correlation between control and salt (*r* = 0.66) but higher correlation with SHS and HS (*r* = 0.80 and 0.85, respectively) indicating that salt induces profound changes in metabolite accumulation in pericarps, which are attenuated by heat. The major effect of heat over salinity has been recently reported in *Arabidopsis thaliana* subjected to single, double and triple combination of heat, salt, and osmotic stresses. However, in this report it was found that heat predominated plant response to a combination of stresses both at the protein and metabolite levels [65].

Notably, isomaltose and cellobiose appear to be up-accumulated under heat treatments (HS and particularly under SHS), but reduced under salt treatment (Table 1), both are disaccharides that act as reducing sugars. Although not much is known about the role played by these sugars in plant growth and development, several reports have demonstrated increased abundance in plant tissues under stress conditions. Accordingly, an increase in isomaltose was reported under water stress conditions [48,66,67] or following exposure to ABA, salt, and cold in the moss *Physcomitrella patens* [68]. Cellobiose is the major product of cellulose hydrolysis by microorganisms and may function as a chemoattractant for motile cellulolytic bacteria such as *Cellulomonas gelida* [69]. Furthermore, in a recent study cellobiose was found to play a role as a “danger” signal that promote plant defenses, which are commonly triggered by microbe-derived elicitors [70]. The sugar alcohol sorbitol implicated in plant stress tolerance was highly up-accumulated in pericarps from salt-treated plants (salt and SHS). Indeed, sorbitol is accumulated to high levels in *Plantago maritima* and might play an adaptive role to conferring salt tolerance [71,72]. Sorbitol also accumulates in the leaves of various species in response to water stress [73,74]. In *Arabidopsis*, sorbitol was accumulated in response to drought stress [75], and *Arabidopsis* mutant in the *SORBITOL DEHYDROGENASE* (*SDH*) gene, whose product convert sorbitol to fructose, conferred tolerance to drought stress [76]. Thus, sorbitol accumulation in pericarps of salt-treated plants could have a beneficial effect on progeny seed germination and growth, particularly under salt and water stress conditions.

The integration of proteins and metabolites accumulated in pericarps of control vs. stress-treated plants revealed a high correlation between the relative content of certain metabolites and the levels of enzymes involved in their biosynthetic pathways, such as the TCA cycle and amino acid synthesis. This correlation indicates that the proteins and metabolites stored in the dead pericarps may reflect their abundance in live pericarps.

The TCA cycle, which occurs in mitochondria is important for the generation of energy in the form of ATP and reducing molecules as well as for generating multiple precursor metabolites required for the synthesis of other molecules such as nucleotides, lipids, and amino acids. TCA intermediates may also function as signaling molecules that control nuclear function [77]. Under single and combined stresses, the relative contents of the TCA intermediates, namely, citrate, succinate, fumarate, and malate were reduced. This is consistent with a recent report addressing the effect of maternal environment on pericarp properties demonstrating a considerable reduction in TCA cycle intermediates in pericarps of the desert plant *A. hierochuntica* following exposure to salt [21]. Furthermore, *Arabidopsis* and Populus plants subjected to heat stress showed significant reduction in TCA cycle intermediates [78,79]. In most cases, exposure to salinity resulted in reduced photosynthesis concomitantly with transient increase in respiration activity until carbon supply is severely depleted [80]. Although there is no clear trend for the response of the TCA enzymes to salinity, summary of published data revealed that about 50% of the TCA enzymes were upregulated in salt-sensitive plants and about 50% were downregulated in salt-tolerant plants [81]. Our proteome data of the dead pericarp showed that all TCA enzymes were up-accumulated under salinity or SHS conditions in parallel with a notable reduction in TCA intermediates. However, such correlation was not observed for pericarps from HS-treated plants where TCA cycle intermediates were downregulated irrespective of their TCA enzyme levels. This outcome can be explained if the TCA cycle is not well fueled by pyruvate as a result of reduction in photosynthesis or alternatively by channeling pyruvate for production of alanine and α-ketoglutarate rather to the production of acetyl CoA, or that α-ketoglutarate produced in the TCA cycle exits the cycle and routed for production of proline. Indeed, it has been reported that prolonged warming and heat shock significantly reduced the conversion of pyruvate into acetyl-CoA [80].

## 5. Conclusions

Exposure of plants during the reproductive stage to single and combined salinity and short episodes of high temperature had an enormous impact on the properties of dead pericarps. Commonly, dead pericarps of *B. juncea* are considered as agricultural waste, yet they appear to function as a rich storage for multiple proteins and metabolites whose levels and composition are changed significantly and variably under single and combined stresses. It appears that salinity has a dominant effect on plant response to combined stresses at the protein level, while heat appears to be the major factor affecting metabolites accumulation in dead pericarps of *B. juncea*. Thus, heat shock and salt differently regulate metabolic profile in the pericarp, with salt acting mainly via regulation of protein abundance, while heat shock often acting irrespective of proteins to control pericarp metabolic profile.

## Figures and Tables

**Figure 1 ijms-22-07076-f001:**
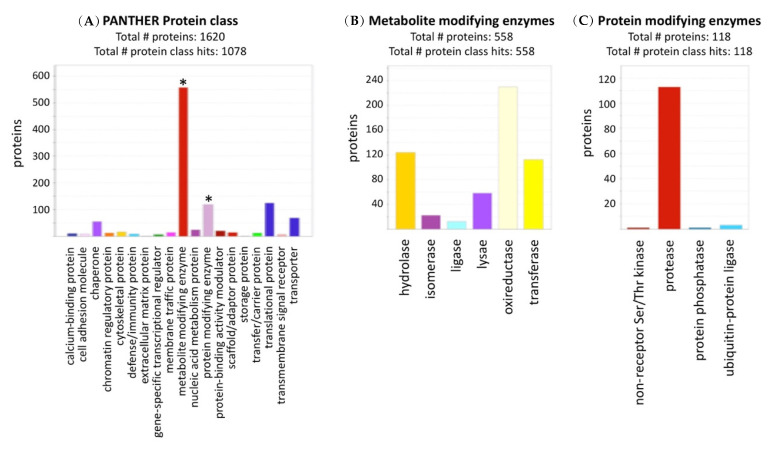
Protein class categorization of proteins accumulated in pericarps derived from treated and untreated plants. (**A**) Protein class categorization highlighting metabolites and protein modifying enzymes among the major groups in this category (asterisks). (**B**) Categorization of metabolite modifying enzymes. (**C**) Categorization of protein modifying enzymes. Note the abundance of proteases in this category. Categorization was performed with PANTHER v.16 [38].

**Figure 2 ijms-22-07076-f002:**
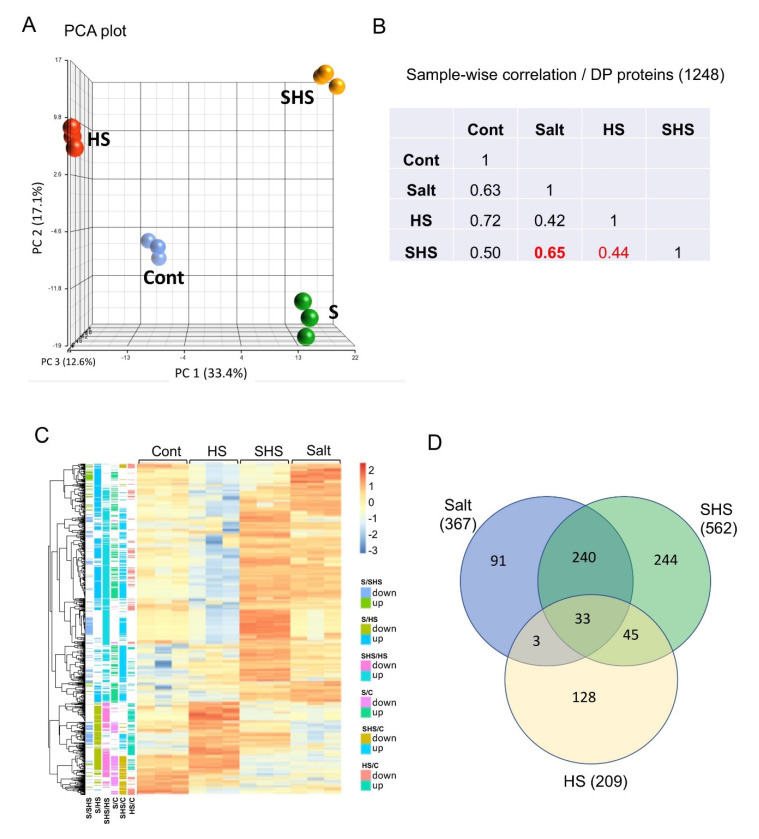
Proteome analysis of pericarps. (**A**) Principal Component Analysis (PCA) score plot comparing the proteome profiles between pericarps derived from control, salt (S), heat shock (HS), and S + HS-treated plants. (**B**) Sample-wise correlation using LFQ values per sample. Red square indicates high correlation between salt (S) and combined S + HS (SHS). (**C**) A hierarchical clustering of DP proteins (n = 1248, passing FDR adjusted *p*-value < 0.05 in any of the pairwise comparisons) identified in pericarps derived from control and stress-treated plants. The color key represents the log 2 values of treatments and control. On the left, up and down accumulated proteins between treatments. Color codes are shown on the right. (**D**) Venn diagram of the differentially present proteins in pericarps. The number in brackets are the number of DP proteins between treatments and control passing the cutoff (FC > 2; *p*-value < 0.05).

**Figure 3 ijms-22-07076-f003:**
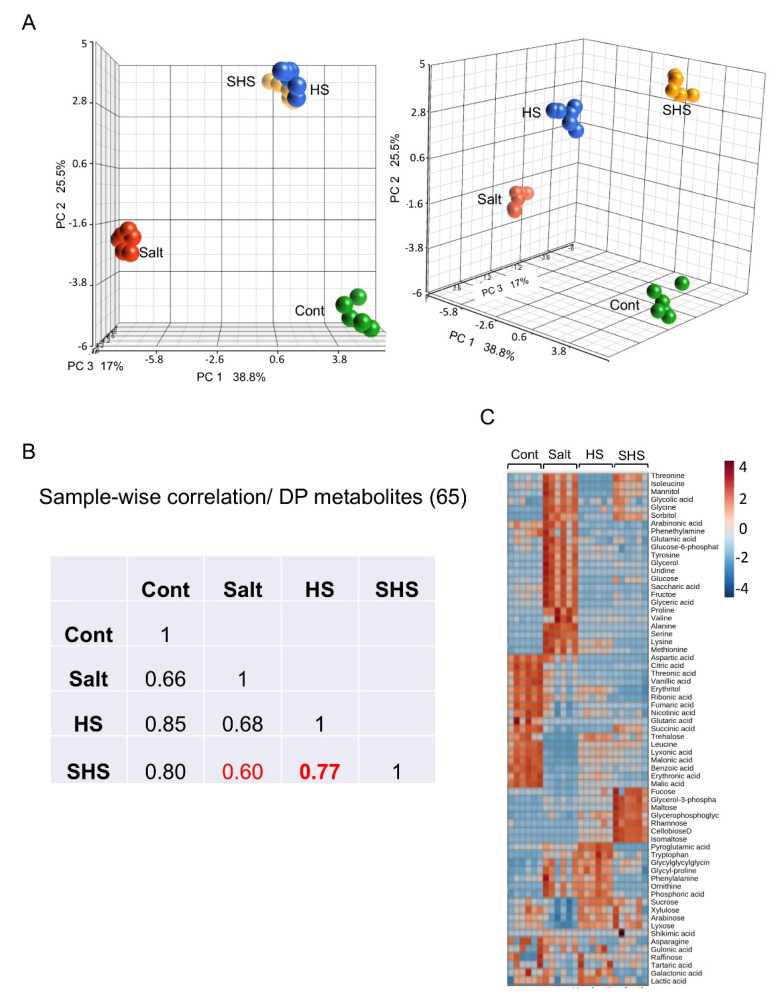
Metabolome analysis of dead pericarps. (**A**) PCA score plot comparing the 65 primary metabolites between pericarps derived from control (Cont), salt (S), heat shock (HS) and SHS-treated plants. (**B**) Sample-wise correlation of DP metabolites showing a relatively high correlation between HS and SHS treatments. (**C**) Hierarchical clustering of differentially present metabolites in dead pericarps after normalization. A comparison of the listed metabolites up and down accumulated in all treatments is shown. Metabolite levels are color-coded with brown and blue representing up and down accumulation, respectively. Each treatment (6 repeats) is represented by six columns.

**Figure 4 ijms-22-07076-f004:**
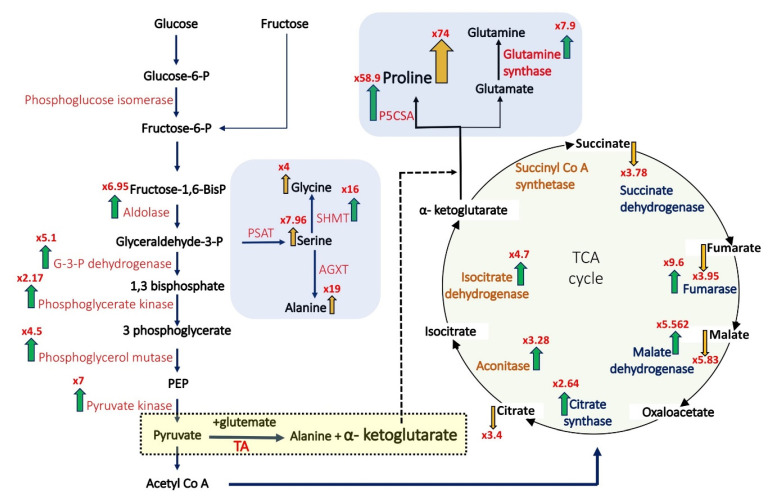
Proteome-metabolome correlation analysis. The metabolites and the enzymes present in pericarp extracts of control and salt-treated *B. juncea* plants involved in the glycolysis pathway, TCA cycle, and amino acids synthesis are shown. Green arrows pointing upward indicate the up-accumulation of enzymes in salt-treated pericarps. Brown arrows pointing upward and downward indicate up-accumulation and down-accumulation of metabolites, respectively. Numbers at the arrow head indicate fold change compared to control pericarps. P5CSA, Delta-1-pyrroline-5-carboxylate synthase A; PSAT, phosphoserine aminotransferase; SHMT, serine hydroxymethyl transferase; AGXT, Alanine glyoxylate aminotransferase; TA, transaminase.

**Table 1 ijms-22-07076-t001:** Fold change (treatments vs. control) in levels of amino acids, TCA cycle intermediates and sugars (*p* value for each comparison is given).

Amino Acids	FC (S/C)	*p*-(S/C)	FC (HS/C)	*p*-(HS/C)	FC (SHS/C)	*p*-(SHS/C)
Alanine	19.2926	4.46 × 10^−11^	2.52061	7.19 × 10^−4^	3.05894	1.026 × 10^−4^
Asparagine	−1.12148	8.06 × 10^−1^	−2.28305	8.82 × 10^−2^	−2.25804	9.23 × 10^−2^
Aspartic acid	−1.94888	3.23 × 10^−10^	−3.7717	9.29 × 10^−16^	−3.8614	6.62 × 10^−16^
Glutamic acid	1.14699	1.59 × 10^−1^	−1.45126	7.60 × 10^−4^	−1.22451	4.33 × 10^−2^
Glycine	4.02944	5.6 × 10^−9^	1.15841	3.2 × 10^−1^	2.79745	6.53 × 10^−7^
Isoleucine	1.22608	1.045 × 10^−4^	−2.6079	9.85 × 10^−16^	1.13231	8.14 × 10^−3^
Leucine	−20.6648	2.47 × 10^−22^	−1.9999	4.23 × 10^−10^	−1.70454	3.38 × 10^−8^
Lysine	11.5683	1.17 × 10^−12^	6.32415	1.97 × 10^−10^	−2.1936	6.81 × 10^−5^
Methionine	4.1542	9.15 × 10^−10^	2.54212	7.8 × 10^−7^	1.53483	4.14 × 10^−3^
Phenylalanine	−1.00857	9.129 × 10^−1^	1.25066	8.79 × 10^−3^	−1.36008	7.17 × 10^−4^
Proline	74.2138	5.01 × 10^−10^	6.90514	6.92 × 10^−5^	2.92048	1.17 × 10^−2^
Serine	7.96858	5.73 × 10^−14^	1.81162	3.92 × 10^−5^	−1.0454	6.99 × 10^−1^
Threonine	1.33931	1.14 × 10^−7^	−1.97091	4.29 × 10^−14^	1.58649	5.3 × 10^−11^
Tryptophan	3.97328	1.17 × 10^−5^	7.35323	5.84 × 10^−8^	2.06643	6.43 × 10^−3^
Tyrosine	4.62252	7.6 × 10^−5^	1.62435	1.31× 10^−1^	−1.97086	4.00 × 10^−2^
Valine	2.73897	6.46 × 10^−4^	1.28322	3.29 × 10^−1^	1.46609	1.41 × 10^−1^
**TCA intermediates**						
Citric acid	−3.46997	8.32 × 10^−17^	−4.02513	9.25 × 10^−18^	−3.06975	6.21 × 10^−16^
Fumaric acid	−3.95859	6.4 × 10^−9^	−2.49273	3.28 × 10^−6^	−2.39499	6.01 × 10^−6^
Malic acid	−5.83132	9.27 × 10^−21^	−2.67643	8.36 × 10^−16^	−2.72549	5.87 × 10^−16^
Succinic acid	−3.7801	1.58 × 10^−17^	−3.25029	1.65 × 10^−16^	−1.40852	5.45 × 10^−7^
**Sugars**						
CellobioseD	−4.05546	1.03 × 10^−14^	14.7441	3.03 × 10^−20^	49.1817	1.98 × 10^−23^
Fructose	1.10372	4.36 × 10^−2^	−1.5159	1.57 × 10^−8^	−1.19213	1.03 × 10^−3^
Glucose	1.83712	2.26 × 10^−4^	−1.82724	2.48 × 10^−4^	1.15139	3.10 × 10^−1^
Isomaltose	−3.84895	8.46 × 10^−4^	7.46703	1.01 × 10^−5^	20.625	2.57 × 10^−8^
Maltose	1.96725	2.26 × 10^−12^	2.53343	5.7 × 10^−15^	10.0327	1.05 × 10^−22^
Mannitol	1.75857	8.02 × 10^−10^	−2.03151	1.43 × 10^−11^	1.59146	2.08 × 10^−8^
Raffinose	1.45761	4.10 × 10^−1^	1.99953	1.37 × 10^−1^	2.05081	1.24 × 10^−1^
Rhamnose	−1.91282	1.48 × 10^−2^	2.26868	3.063 × 10^−3^	7.01187	1.15 × 10^−7^
Sorbitol	17.897	4.48 × 10^−10^	4.11097	2.23 × 10^−5^	18.2656	3.97 × 10^−10^
Sucrose	−1.89317	8.65 × 10^−10^	1.05925	3.42 × 10^−1^	1.05054	4.14 × 10^−1^
Trehalose	−6.46999	3.69 × 10^−10^	−1.2477	1.94 × 10^−1^	−1.80302	1.87 × 10^−3^

**Table 2 ijms-22-07076-t002:** Relationship between proteins and metabolites. Fold change (FC) ratio of proteins and metabolites between treatments and control (*p* value < 0.05). P, protein; M, metabolite. No correlation between protein and metabolite levels is colored red.

	Affected	Salt/Control (FC)		HS/Control (FC)		SHS/Control (FC)	
Proteins/ID	Metabolite	P	M	P	M	P	M
**TCA cycle**							
Aconitase/M4DTK2	Citrate	3.27	−3.469	nc	−4.02	3.29	−3.07
Fumarate hydratase/M4CJR6	Fumarate	9.98	−3.9	nc	−2.49	6.925	−2.39
Malate dehydrogenase/M4FEZ0	Malate	5.56	−5.83	7.757	−2.67	15.41	−2.72
**Amino acids**							
P5CSA/M4CLC5	Proline	58.9	74.2	−2.343	6.9	33.39	2.92
SHMT/M4D5N9	Glycine	16	4	nc	nc	12.07	2.79

P5CSA, Delta-1-pyrroline-5-carboxylate synthase A; SHMT, serine hydroxymethyl transferase.

## Data Availability

The data that support the findings of this study are available in the main text and in Appendix A.

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
