# Peer review of "Differential Response to Single and Combined Salt and Heat Stresses: Impact on Accumulation of Proteins and Metabolites in Dead Pericarps of Brassica juncea"

_ijms, 2021, doi:10.3390/ijms22137076_

Round 1
Reviewer 1 Report
In this manuscript, the author studies the differential response to single and combined salt and heat stresses and their impact on the accumulation of proteins and metabolites in dead pericarps of Brassica juncea. The author evaluated the effect of salinity (S), short episodes of high temperature (HS), and combination of S+HS (SHS), at the reproductive phase, on the properties of dead pericarps of Brassica juncea. Proteome and metabolome analyses revealed multiple proteins and metabolites stored in dead pericarps whose levels and composition were altered under single and combined stress conditions. The protein profile of SHS showed a higher correlation with salt than with HS, indicating the dominant effect of salt over heat stress. On the other hand, the analysis of metabolites showed that the profile of SHS has a better correlation with HS than with salt. The integration of metabolic and proteomic data showed that changes in TCA cycle intermediates and certain amino acids (e.g., proline) under salt treatments (S and SHS) are highly correlated with changes in proteins involved in their biosynthetic pathways. Thus, the accumulation of proteins and metabolites in dead pericarps is differently affected by single and combination of salt and heat stresses. Salinity appears to dominate plant response to combined stresses at the protein level, while heat appears to be the significant factor affecting metabolite accumulation in dead pericarps.
The manuscript is very well written; however, for the betterment of the manuscript, I have few suggestions given below:
Change at
L17 showed higher correlation to showed a higher correlation.
L17 over heat to overheat.
L42, L286, heat waves to heatwaves.
L43 Choose a different word from occur.
L45 reproductive stages has to reproductive stages have.
L47 plants to combination of to plants to a combination of.
L157 oxireductases to oxidoreductases.
L161 principle to principal.
L172 demonstrates to demonstrate.
L204 that belong to that belongs.
L208 supplementary to Supplementary.
L214 Division Control protein to Division Control Protein.
L231 has significant effect to has a significant effect.
L235 SHS has higher correlation to SHS has a higher correlation.
L287 serious threat on global to serious threat to global.
L289 that combination to that a combination,
L293 stresses to stresse.
L295 recent published data to recently published data.
L301 populations of Avena sterilis is affected to populations of Avena sterilis are affected.
L309 showed better correlation to showed a better correlation.
L338 that confer better resistance to that confers better resistance.
L360, L368 combination of stresses to a combination of stresses.
L383 adaptive role conferring salt tolerance to adaptive role to conferring salt tolerance.
L388 have beneficial effect on progeny seeds germination to have a beneficial effect on progeny seeds germination.
L402 demonstrating considerable reduction to demonstrating a considerable reduction.
L427 has dominant effect on plant to has a dominant effect on plant.
Author Response
Thanks for commenting on the manuscript.
L17 showed higher correlation to showed a higher correlation.
Changed as suggested [line 24]
L17 over heat to overheat.
No change! Over in the sense of ‘greater than’
L42, L286, heat waves to heatwaves.
Heat waves were changed to heatwaves throughout the text.
L43 Choose a different word from occur.
Changed to ‘appear’ line 58
L45 reproductive stages has to reproductive stages have.
Chenged! Line 60
L47 plants to combination of to plants to a combination of.
Changed! Line 62
L157 oxireductases to oxidoreductases.
Changed! Line 186
L161 principle to principal.
Changed Line 190
L172 demonstrates to demonstrate.
No Changed, demonstrates is related to Hierarchical clustering
L204 that belong to that belongs.
Changed, Line 234
L208 supplementary to Supplementary.
Changed! Line 237
L214 Division Control protein to Division Control Protein.
Changed! Line 242
L231 has significant effect to has a significant effect.
Changed Line 262
L235 SHS has higher correlation to SHS has a higher correlation.
Changed! Line 270
L287 serious threat on global to serious threat to global.
Changed! Line 334
L289 that combination to that a combination,
Changed! Line 335
L293 stresses to stresse.
Changed to stress, line 339
L295 recent published data to recently published data.
Changed! Line 340
L301 populations of Avena sterilis is affected to populations of Avena sterilis are affected.
Changed! Line 346
L309 showed better correlation to showed a better correlation.
Changed! Line 355
L338 that confer better resistance to that confers better resistance.
Changed! Line 383
L360, L368 combination of stresses to a combination of stresses.
Changed Line 412
L383 adaptive role conferring salt tolerance to adaptive role to conferring salt tolerance.
Changed Line 425
L388 have beneficial effect on progeny seeds germination to have a beneficial effect on progeny seeds germination.
Changed Line 430
L402 demonstrating considerable reduction to demonstrating a considerable reduction.
Changed Line 444
L427 has dominant effect on plant to has a dominant effect on plant.
Changed Line 468
Reviewer 2 Report
The work entitled "Differential response to single and combined salt and heat stresses: impact on accumulation of proteins and metabolites in 2 dead pericarps of Brassica juncea" presents an extensive proteomic and metabolic analysis of B. juncea under different stresses, in order to characterize the specific responses to a combination of stresses. The work is original and it is performed in a robust way, ensuring the quality of the experimental work performed. Despite this, I consider that some points of the current manuscript must be improved in order to provide a manuscript detailed enough to be useful for the readers.
- In material and methods, the analysis of the samples are not detailed in a proper way. In line100, the technique, equipment and analysis is not detailed, since it is in the results section when it is stated that it has been analyzed by LC-MS/MS (Line 150). Please, improve the description of all the section, since this is only one example.
- All the packages and bioinformatic resources need to be referenced, they are as important as experimental protocols.
- When a PCA analysis is performed, the elements defined as PC1 and PC2 need to be specified, specially if the weight of each PC is going to be used to bring some conclusions.
- In line 259, it is stated "well correlated". Which statistic is used to affirm a good correlation? If none, I suggest to use other way to express this idea.
- In line 368, gene expression is mentioned. Is it a mistake? Gene expression has not been studied in this version of the manuscript.
- And, finally, my main concern is the fact that, although there is an exhaustive description of the effects of S, HS and SHS; the main idea, that was to find the unique response to a combination of stress is not developed. I would like to have more information of the proteins/metabolites that are only present in SHS and not in the other groups.
Author Response
- In material and methods, the analysis of the samples are not detailed in a proper way. In line100, the technique, equipment and analysis is not detailed, since it is in the results section when it is stated that it has been analyzed by LC-MS/MS (Line 150). Please, improve the description of all the section, since this is only one example.
The methodology aspect (LC-MS/MS) was omitted from the result section and the reader is directed to the Web site of Smoler Protein Research Center at the Technion, Israel for Details regarding the proteomic instrumentation.
[See M&M subsection 2.2 Proteome analysis, lines 124-126]
- All the packages and bioinformatic resources need to be referenced, they are as important as experimental protocols.
- References to packages (Partek and R) resources were added. [see lines 130-133]
- When a PCA analysis is performed, the elements defined as PC1 and PC2 need to be specified, specially if the weight of each PC is going to be used to bring some conclusions.
- The metabolites most contributing to PC1 and PC2 were added.[lines 266-269]
- In line 259, it is stated "well correlated". Which statistic is used to affirm a good correlation? If none, I suggest to use other way to express this idea.
- Well correlated was changed to ‘is associated with’ [line 298]
- In line 368, gene expression is mentioned. Is it a mistake? Gene expression has not been studied in this version of the manuscript.
- Changed to ‘protein’ [line 412]
- And, finally, my main concern is the fact that, although there is an exhaustive description of the effects of S, HS and SHS; the main idea, that was to find the unique response to a combination of stress is not developed. I would like to have more information of the proteins/metabolites that are only present in SHS and not in the other groups.
- The information exists for proteins (e.g., thionin) and metabolites (cellobiose and isomaltose) highly accumulated in SHS are given and we believe we provided the most available data regarding these substances. [see lines 369-383]
Reviewer 3 Report
Dear Authors
The present manuscript entitled "Differential response to single and combined salt and heat stresses: impact on accumulation of proteins and metabolites in dead pericarps of Brassica juncea" demonstrated plant response to combined stresses at the primary metabolites and protein level, while heat appears to be the major factor affecting metabolite accumulation in dead pericarps of mustard plants. The manuscript is very well organized and presented, especially Correlation drawn between metabolites and proteins is impressive. There are some very small queries regarding seed material and stress conditions, please find it below
- Please specify the type of mustard seeds, for example it is oriental mustard, chinese mustard or indian mustard as they may have different responses in different climatic conditions. Any other details of mustard seeds used in present study would be additional for readers.
- Please provide the details of water used in irrigation of pots (line 85-86, line 91-92).
- Please explain the basis of 3 intervals of heat shock treatment (37oC, 3 h each) in a course of 5 days.
Thank you
Regards
Author Response
- Please specify the type of mustard seeds, for example it is oriental mustard, chinese mustard or Indian mustard as they may have different responses in different climatic conditions. Any other details of mustard seeds used in present study would be additional for readers.
Mustard seeds were imported from India (indicated in M&M, line 98).
- Please provide the details of water used in irrigation of pots (line 85-86, line 91-92).
Details regarding water use are given [M&M subsection 2.1_.
- Please explain the basis of 3 intervals of heat shock treatment (37oC, 3 h each) in a course of 5 days.
The reasoning for 3 intervals of HS (37oC) treatment is based on the World Meteorological Organization defining a heat wave as 5 or more consecutive days of prolonged heat in which the daily maximum temperature is higher than the average maximum temperature by 5 °C (9 °F) or more. Thus, the heat wave temperature ranges from 30oC in the Netherlands to over 40oC (for at least 3 consecutive days) in Adelaide, South Australia. [added to subsection 2.1]
Round 2
Reviewer 1 Report
I am happy with the author's reply. The manuscript can be accepted in its current format.